# Rise and Fall of SARS-CoV-2 Lineage A.27 in Germany

**DOI:** 10.3390/v13081491

**Published:** 2021-07-29

**Authors:** Sébastien Calvignac-Spencer, Matthias Budt, Matthew Huska, Hugues Richard, Luca Leipold, Linus Grabenhenrich, Torsten Semmler, Max von Kleist, Stefan Kröger, Thorsten Wolff, Martin Hölzer

**Affiliations:** 1Epidemiology of Highly Pathogenic Microorganisms, Robert Koch Institute, 13353 Berlin, Germany; 2Influenza and other Respiratory Viruses, Robert Koch Institute, 13353 Berlin, Germany; BudtM@rki.de (M.B.); WolffT@rki.de (T.W.); 3Methodology and Research Infrastructure, Bioinformatics, Robert Koch Institute, 13353 Berlin, Germany; HuskaM@rki.de (M.H.); RichardH@rki.de (H.R.); 4Methodology and Research Infrastructure, Information and Research Data Management, Robert Koch Institute, 13353 Berlin, Germany; LeipoldL@rki.de (L.L.); GrabenhenrichL@rki.de (L.G.); 5Methodology and Research Infrastructure, Genome Sequencing and Genomic Epidemiology, Robert Koch Institute, 13353 Berlin, Germany; SemmlerT@rki.de; 6Systems Medicine of Infectious Disease, Robert Koch Institute, 13353 Berlin, Germany; KleistM@rki.de; 7Infectious Disease Epidemiology, Robert Koch Institute, 13353 Berlin, Germany; KroegerS@rki.de

**Keywords:** public health surveillance, SARS-CoV-2 variants, viral genomes, molecular sequence data

## Abstract

Here, we report on the increasing frequency of the SARS-CoV-2 lineage A.27 in Germany during the first months of 2021. Genomic surveillance identified 710 A.27 genomes in Germany as of 2 May 2021, with a vast majority identified in laboratories from a single German state (Baden-Wuerttemberg, *n* = 572; 80.5%). Baden-Wuerttemberg is located near the border with France, from where most A.27 sequences were entered into public databases until May 2021. The first appearance of this lineage based on sequencing in a laboratory in Baden-Wuerttemberg can be dated to early January ’21. From then on, the relative abundance of A.27 increased until the end of February but has since declined—meanwhile, the abundance of B.1.1.7 increased in the region. The A.27 lineage shows a mutational pattern typical of VOIs/VOCs, including an accumulation of amino acid substitutions in the Spike glycoprotein. Among those, L18F, L452R and N501Y are located in the epitope regions of the N-terminal- (NTD) or receptor binding domain (RBD) and have been suggested to result in immune escape and higher transmissibility. In addition, A.27 does not show the D614G mutation typical for all VOIs/VOCs from the B lineage. Overall, A.27 should continue to be monitored nationally and internationally, even though the observed trend in Germany was initially displaced by B.1.1.7 (Alpha), while now B.1.617.2 (Delta) is on the rise.

## 1. Introduction

From the beginning of last year, the world has witnessed the emergence and unprecedented spread of SARS-CoV-2 which has so far claimed at least 3.5 million lives worldwide (June 2021, WHO). Viral genome sequencing and subsequent integrated data analyses proved to be essential [1,2,3,4] to track the spread of the virus and to detect emerging mutations as well as variants of concern/interest (VOC/I) that can be associated with antibody escape or higher transmissibility [5,6,7,8].

Since January 2021, the German Electronic Sequence Data Hub (DESH) has been active, through which all sequencing laboratories in Germany transmit SARS-CoV-2 genome sequences to the Robert Koch Institute (RKI), Germany’s national public health institute. This technical platform enables the pooling of sequence data throughout Germany, further enriched by in-house-sequenced samples as part of the SARS-CoV-2 Integrated Molecular Surveillance lab-network (IMS-SC2). Data submitted to DESH from sequencing laboratories across the country, as well as IMS-SC2 sequence data, in combination with additional international data from resources such as GISAID [9] or EMBL-EBIs COVID-19 Data Portal [10], enabled comprehensive genomic surveillance of the pandemic across Germany.

SARS-CoV-2 genomes belonging to the A.27 lineage were already identified in France in mid-January 2021 and analyzed in the context of other sequences available at that time, originating from the Comoros archipelago, Western European countries (mainly metropolitan France), Turkey and Nigeria [11]. In addition, colleagues from the Rhein-Neckar district in Germany reported a local occurrence and decline of the A.27 variant in January and March 2021 [12], while Anoh et al. also found A.27 sequences on the rise in western sub-Saharan Africa between May 2020 and March 2021 [13]. Here, we now extend these analyses and also use the example of the spread of A.27 in Germany in the first months of 2021 to show how genomic surveillance enables monitoring of SARS-CoV-2 variants at the national level. A.27 is of particular interest because the lineage shows a mutational pattern typical of VOIs/VOCs, including an accumulation of amino acid changes in the spike that are thought to lead to immune escape and higher transmissibility.

## 2. Materials and Methods

### 2.1. Genome Reconstruction

In addition to the sequences obtained via DESH, three samples assigned to the A.27 lineage were sequenced directly at the RKI as part of the IMS-SC2 lab-network. Amplicon sequencing was either performed via Illumina (*n* = 1) or Nanopore (*n* = 2) and consensus genomes were reconstructed using covPipe (Illumina data, unpublished, v3.0.1, https://gitlab.com/RKIBioinformaticsPipelines/ncov_minipipe, accessed on 1 May 2021) or poreCov (Nanopore data, [14], v0.7.8, https://github.com/replikation/poreCov, accessed on 1 May 2021). For Illumina, amplicon sequencing was performed using the CleanPlex SARS-Cov-2 Flex amplicon panel on an iSeq 100 system with 150 bp paired-end reads yielding 283k reads for this sample. Nanopore sequencing was performed using the ARTIC V3 primer set on a MinION flow cell resulting in 116k and 108k reads per A.27 sample, respectively. Both sequencing and reconstruction approaches resulted in high-quality consensus sequences with an N content of 0.40 % (both Nanopore-derived sequences) and 1.64 % (Illumina) per genome.

### 2.2. Detecting Increase in Proportion

Based on the genomic data, we tested for differences in the proportion of the A.27 lineage using a Fisher exact test. Tests were performed separately for suspect and random sampling strategies. For each German state, the test was performed on a 2 × 2 count table showing, for pairs of consecutive calendar weeks (CW), the number of A.27 samples and the total number of non-A.27 samples in a state. If no A.27 sequences were identified for a particular federal state in a given week, that week was skipped and sequences from the next week for that state were considered instead. We chose this approach to be more conservative in detecting an increase in the proportion. Only states in which A.27 samples were detected in at least three CW were considered. The obtained set of *p*-values was then corrected for multiple testing using a Benjamin-Hochberg procedure, keeping the adjusted *p*-values below 0.1.

### 2.3. Phylogenetic Reconstruction

We used all available A.27 genomes and the selection of A genomes from NextStrain [15] global build to assemble a dataset for phylogenetic analyses including 718 A.27 genomes from RKI and GISAID. After excluding low quality/missing sampling date genomes (<90 % reference genome coverage, >10 non-ACGTN ambiguous positions, >40 SNPs with respect to NC_045512.2), we aligned a total of 875 genomes with MAFFT using default parameters [16]. We used the resulting alignment to estimate a no-clock maximum-likelihood tree with IQ-TREE2 [17]. This tree showed a strong temporal signal in a regression of root-to-tip distance versus time (R${}^{2}$: 0.81) and was therefore used to estimate a time tree using tip dates only with TreeTime [18].

### 2.4. Three-Dimensional Structure of the S Protein

The PyMOL software version 1.7.2.1 [19] was used to label variant residues on the structure of the spike protein based on the template structure 7kj2 obtained from the Protein Data Base (www.pdb.org, accessed on 1 July 2021) and provided by [20].

## 3. Results and Discussion

### 3.1. Increase of A.27 Cases in Baden-Wuerttemberg, Germany

A total of 377 A.27 genomes from 16 different (mostly European) countries have been made available on GISAID (as of 10 May 2021). A.27 was first detected in Denmark in mid-December 2020 but rather 214 genomes were reported from France including 12 from Mayotte, an overseas region and single territorial collectivity of France. The numbers might therefore point to an origin of A.27 in France where the lineage may have spread early. However, A.27 was also found in Côte d’Ivoire (western sub-Saharan Africa) [13] and Burkina Faso and Togo (based on GISAID sequences) in early 2021, indicating another possible origin of A.27 in West Africa. In addition, A.27 cases have been reported from West African countries based on sequencing data from Belgian forces involved in overseas operations. [21].

Genomic surveillance at the RKI identified 710 sequences belonging to the lineage A.27 (as of 2 May 2021). The earliest sequence was discovered in a laboratory in Baden-Wuerttemberg (BW) and occurred in calendar week (CW) 01 in 2021; since then, a vast majority of A.27 cases in Germany were sequenced in BW (*n* = 572 to CW16, based on data through 2 May 2021). The relative abundance of this lineage in the region increased until CW08 (6.12 %) but has since then decreased (1.21 % in CW13). Meanwhile, the frequency of the VOC B.1.1.7 kept increasing in the region (5.84 % in CW 03, 71.0 % in CW 14; see corresponding reports at https://www.rki.de/DE/Content/InfAZ/N/Neuartiges_Coronavirus/DESH/Berichte-VOC-tab.html, accessed on 27 July 2021).

Of the 710 A.27 genomes from Germany, 206 were obtained following a random sampling strategy, while 271 were collected as ‘suspect samples’ based on variant-specific PCRs or epidemiological circumstances (remainder unknown). For sequences from both categories (‘suspect’, ‘random’), the proportion of reported A.27 sequences was compared between calendar weeks and significant increases were detected (Fisher exact test, adjusted *p*-value < 0.1, see Methods). In CW 07–08, we observed a 2.1-fold increase in proportion among the suspect samples in the BW region, followed the next week by a 3-fold increase on the set of samples obtained using a random sampling strategy (Figure 1A). Interestingly, on the random samples and during the same period CW 07–10, Schleswig-Holstein also showed an impressive 33-fold increase in proportion (Figure 1A) but absolute numbers were still low (Table 1). Note that no A.27 sequences were detected in CW08 and 09 for that region.

### 3.2. Epidemiological Data

For 205 cases reported between CW03 and 14, additional information was available via the national electronic reporting system for surveillance of notifiable infectious diseases (SurvNet, implemented in 2001; [22]) (Figure 1B,C; geographic distribution similar to that seen in the entire dataset, data not shown). Most cases were reported for middle-aged patients (Figure 1B). Hospitalization was reported in 17 patients, and three died. None of the 205 reported cases with epidemiologic data originated in Schleswig-Holstein.

### 3.3. Phylogenetic Analysis of A.27 Genomes

For phylogenetic analysis, we assembled a dataset of 718 A.27 genomes from RKI and GISAID and a selection of additional A genomes from NextStrain [15] as an outgroup. All A.27 genomes appeared in a monophyletic group whose most recent common ancestor (MRCA) was dated to early August 2020 (6 August 2020; Figure 2). However, all but three basal A.27 sequences formed a clade whose time to MRCA was much more recent, around late October 2020 (29 October 2020). The tree shows a basal polytomy from which different clades descend. One clade comprises many sequences originating in France, whereas the largest clade mainly comprises German sequences (Figure 2).

### 3.4. Spike Mutations of Interest and Concern in A.27

The A.27 lineage is characterized by a high number of characteristic (lineage-defining) mutations that are also known from other VOC/VOI (Figure 3C). In total, A.27 harbors 17 lineage-defining mutations (Figure 3A and Figure 4A), seven of which result in non-synonymous nucleotide substitutions in the spike protein: L18F, L452R, N501Y, A653V, H655Y, D796Y, G1219V. Among those, three alterations are of particular concern as they are located in the NTD or RBD epitope regions, respectively, and are suspected to result in immune escape (L18F, L452R and N501Y) and/or higher infectivity and transmissibility (L452R and N501Y). Antibody escape data for the RBD, integrating multiple experimental studies [5] (https://jbloomlab.github.io/SARS2_RBD_Ab_escape_maps, accessed on 3 May 2021), assign maximum mutation escape scores of 0.97 (L452R) and 0.90 (N501Y) over multiple antibodies/sera or antibody/serum types.

The polymorphic positions are labeled on the S protein structure (Figure 3B), indicating the localization of L452R (reduced antibody neutralization [7,8]) and N501Y (related to increased transmissibility [6]) in the RBD, A653V/H655Y in proximity to the S1/S2 furin cleavage site at position 681 that promotes infection and cell-cell fusion [23], and D795Y, closely located to the essential TMPRSS2-cleavage site at position 815 [24].

The L18F replacement is part of an antigenic supersite in the N-terminal domain of the spike protein [25]. Antibodies targeting NTD were shown to be less abundant, but more potent than those targeting RBD [25]. Mutations at this position were selected via in vitro passaging experiments and showed cross-resistance to monoclonal antibodies targeting NTD [25]. The S-L18F alteration also appears in VOCs that are associated with immune escape and reinfection, such as B.1.351 and P.1. Recently, the L452R change in the RBD also known from B.1.427/429 [26] and B.1.617.2 and its novel sublineages AY.1 and AY.2 (Figure 3C) was suggested to contribute to escaping human leukocyte antigen-restricted cellular immunity while also increasing the binding affinity to ACE2, thus increasing viral infectivity and potentially enhancing virus replication [27]. S-N501Y also appears in VOCs B.1.1.7, P.1 and B.1.351, and may increase infectivity in vitro [28] and appears to confer resistance against some RBD targeting antibodies (class 1) [29]. The H655Y substitution adjacent to the S1/S2 cleavage site and also known from P.1 was recently detected in a potential new VOI within the A lineage (temporarily designated A.VOI.V2) identified from three cases of incoming travelers from Tanzania to Angola [30].

Importantly, A.27 does not possess the D614G change, which emerged early in the pandemic and rapidly became dominant. D614G is associated with increased receptor binding [31] and infectivity [32]. Neither does A.27 contain the likely functionally equivalent Q613H that is present in A.23.1 [33] (Figure 3C).

Almost all A.27 genomes have multiple deletions such as ORF3a:del258/259 and ORF8:del119/120, a few of which are highly conserved (Figure 3 and Figure 4). We also observed in 58 (8.07 %) of the investigated A.27 genomes a 12 nt long insertion (CTTTCGATCTCT) located at position 27,373 near the 3’-end of ORF6. However, all sequences with this insertion were submitted by a single laboratory via DESH and further investigation showed that the inserted sequence is similar to a potential primer sequence in the first genomic amplicon. Therefore, a technical error in the sequencing protocol or in the genome reconstruction pipeline of the submitting laboratory cannot be excluded.

### 3.5. Spike Mutations of Interest and Concern in Basal A.27 Lineages

The rare, relatively deep-branching lineages in the phylogenetic tree (Figure 2) prompted us to check the mutation profile of the corresponding three genomes (Figure 4, GISAID accessions: EPI_ISL_1170076, EPI_ISL_1567985, EPI_ISL_1353586). None possessed all the seven aforementioned non-synonymous mutations in spike, nor any of the frequently observed deletions (Figure 4). Intriguingly, one of the genomes (EPI_ISL_1567985) encoded S-L452R, S-N501Y, and S-G1219V associated with six other amino acid changes in the spike (A570D, D614G, P681H, T716I, S982A, I1221V), most interestingly comprising D614G and also P681H (Figure 4B). While we report here on these three unusual A.27 genomes, further analysis is required also on their raw sequencing data to confirm the observed changes that distinguish these sequences from standard A.27.

## 4. Conclusions

Here, we report the increasing frequency of the SARS-CoV-2 lineage A.27 in Germany, particularly in the federal state of Baden-Wuerttemberg (BW), in January and February 2021. The origin of A.27 is still unclear. In the context of genomic surveillance in Côte d’Ivoire (western sub-Saharan Africa) between May 2020 and March 2021, Anoh *et al*. also detected cases of A.27 sequences in early 2021 [13]. To investigate Africa as a possible origin, we searched the complete GISAID database (downloaded on 20 July 2021) for sequences submitted from African countries since 1 December 2020 and assigned to A.27. While the number of submitted sequences from Africa is still low, we found hits for A.27 in eleven African countries, eight of which are in West Africa. Among them, Togo (21.87 %), Burkina Faso (8.82 %), and Côte d’Ivoire (4.83 %), as well as Tunisia (12.96 %, North Africa) showed higher prevalence of A.27 which is in line with results from Pirnay *et al*. First cases of A.27 sequences were detected 21 December 2020 in Togo, 22 December 2020 in Burkina Faso, and 2 February 2021 in Tunisia, which fits the time period we observed A.27 in Germany if we assume an earlier origin in West Africa. Thus, we speculate that West Africa may be the origin of the A.27 lineage, which may also explain why early and higher numbers of this specific lineage were first discovered in France and Mayotte. However, we must consider sparse sampling strategies and other caveats (random vs. suspect sampling, travel history, sequencing capacity, overall reporting, and data availability). Therefore, further analyses are needed to fully elucidate the origin of the SARS-CoV-2 A.27 lineage.

Since A.27 shows mutations associated with both higher transmissibility and reduced antibody reactivity, we suggest that enhanced monitoring is warranted on both national and international levels. The A.27 lineage was indeed continuously detected in other European countries (France, Slovenia and the United Kingdom). However, the observed trend in BW clearly showed that A.27 was initially displaced by B.1.1.7 (Alpha) while now B.1.617.2 (Delta) is on the rise and should be monitored closely. Based on genomic data obtained via a random sampling strategy, we also found a significant increase in the proportion of A.27 sequences between CW 07-10 in the federal state of Schleswig-Holstein. Finally, our analyses identified rare but clearly divergent genomes assigned to A.27, suggesting the current definition of the lineage probably needs to be reexamined. One of these rare genomes combines a D614G background and several mutations of concern and should therefore also be monitored.

## Figures and Tables

**Figure 1 viruses-13-01491-f001:**
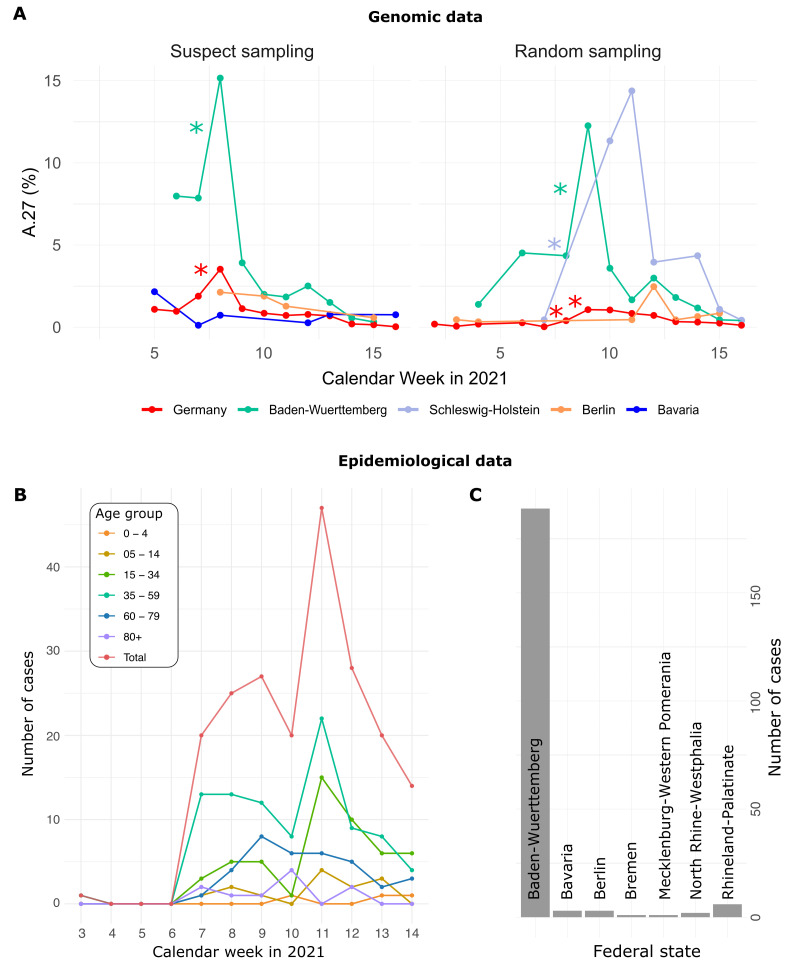
(**A**) Proportion of reported A.27 sequences compared between calendar weeks (CW) and based on suspect (*n* = 271) and random sampling (*n* = 206) strategies. Significant increases in proportion are marked with a star (adjusted *p*-value < 0.1). In CW 07–08, we observed a 2.1-fold increase in the proportion among suspect samples in the BW region and in CW 08–09 among random samples in the same region. Between CW07 and CW10 we observed an impressive 33-fold increase in proportion in Schleswig-Holstein among the random samples, however, no A.27 sequences were detected in CW08 and 09 for that region, and absolute numbers are low. (**B**) Epidemiological data of reported cases of lineage A.27 (*n* = 205) in Germany per age group. Median age of cases was 45 years. Note that information was not available in the reporting system for all data points, so they may have no value. In general, no information was available for CW04–06 as of 30 April 2021. (**C**) Distribution of cases over federal states based on epidemiological data, 92 % were notified in Baden-Wuerttemberg (date of reporting: 30 April 2021). Epidemiological data was not available for all federal states.

**Figure 2 viruses-13-01491-f002:**
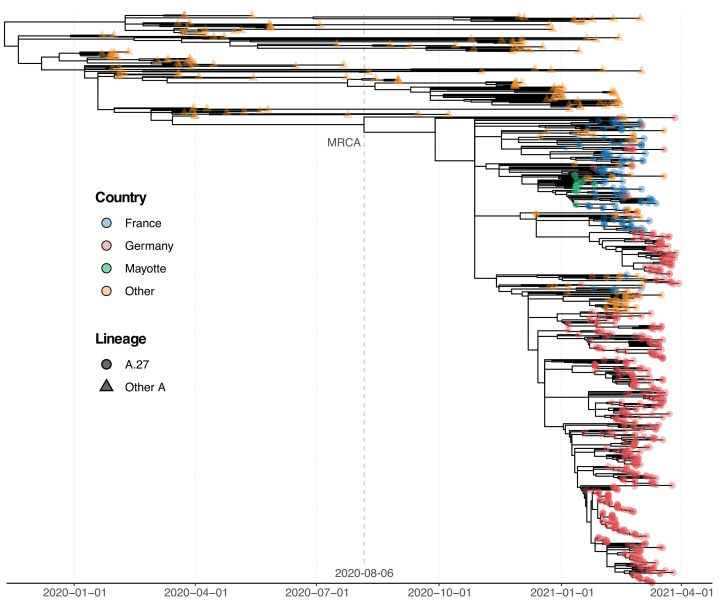
Phylogenetic analysis of A.27 genomes forming a monophyletic group. We also observed three basal A.27 genomes missing some of the seven characteristic A.27 spike mutations and showing no deletions in spike. The most recent common ancestor (MRCA) was dated to 8 August 2020.

**Figure 3 viruses-13-01491-f003:**
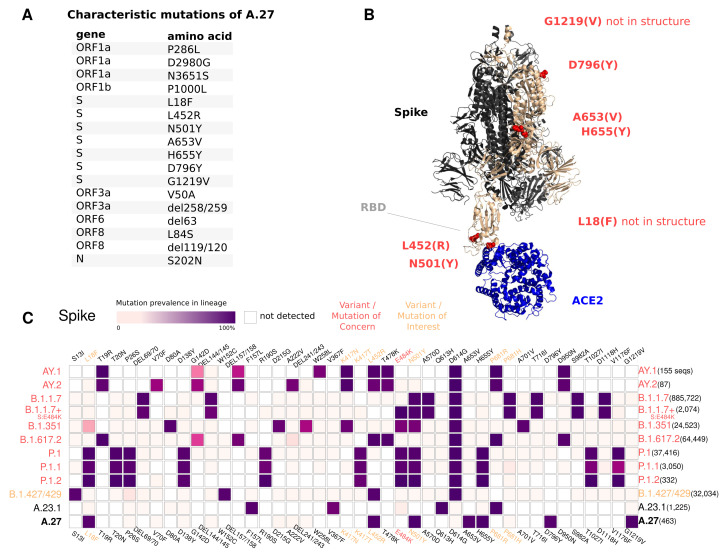
(**A**) Characteristic (lineage-defining) mutations of A.27 derived from https://outbreak.info. (**B**) The trimeric spike protein of SARS-CoV-2 (PDB: 7kj2, [19]) is depicted in complex with its cellular receptor ACE2. The variant residues of lineage A.27 are labeled on one subunit with the A.27 amino acids indicated in parentheses. RBD—receptor binding domain. (**C**) Mutation profiles of selected VOC/VOI in comparison to A.27 and A.23.1 obtained from https://outbreak.info (accessed on 23 June 2021) [34].

**Figure 4 viruses-13-01491-f004:**
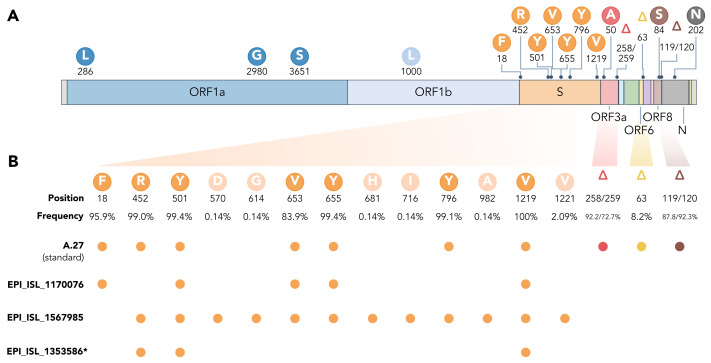
(**A**) Characteristic mutations leading to 17 amino acid alterations or deletions in A.27 genomes with a prevalence of at least 75 %. Figure obtained and adjusted from [35]. (**B**) Overview of spike substitutions and high-frequency deletions in ORF3a, ORF6, and ORF8 in genomes assigned the A.27 lineage via pangolin. Frequencies are calculated based on the set of 718 A.27 genomes as used for the tree in Figure 2. Among “standard” A.27 genomes harboring all seven spike mutations and the three prevalent deletions, we also found three genomes that show a varying mutation profile in the spike and missing all three high-frequency deletions. While the deletion on ORF6 at position 63 is listed with a high frequency (higher 75 %) on outbreak.info (accessed on 24 May 2021), we only observe a frequency of 8.2 % in our data set based on a detection via NextStrain. * EPI_ISL_1353586 is not yet available on GISAID due to rejection of the sequence because of a frameshift.

**Table 1 viruses-13-01491-t001:** The number of detected A.27 whole-genome sequences by calendar week (CW) until 2 May 2021 that were either submitted to the RKI via DESH (*n* = 707) or directly sequenced at the RKI as part of the IMS-SC2 (*n* = 3). Data from a random and suspect sampling strategy are combined. Relative frequencies (in %) in relation to all sequences from the respective CW are given in parentheses. Later time periods may be biased due to missing data. GER—All of Germany, BW—federal state of Baden-Wuerttemberg, SW—federal state of Schleswig-Holstein.

	CW01	02	03	04	05	06	07	08	09	10	11	12	13	14	15	16	2021
GER	3	2	7	29	40	53	61	96	79	84	86	65	52	27	18	8	**710**
	(1.07)	(0.22)	(0.32)	(0.82)	(0.92)	(0.89)	(0.91)	(1.35)	(1.02)	(0.91)	(0.81)	(0.62)	(0.48)	(0.25)	(0.14)	(0.10)	
BW	3	0	6	28	35	52	52	83	76	54	53	51	47	21	9	2	**572**
	(2.75)	(0.00)	(1.34)	(4.21)	(4.69)	(4.47)	(4.12)	(6.12)	(3.98)	(2.06)	(1.72)	(1.64)	(1.21)	(0.59)	(0.29)	(0.22)	
SH	0	0	0	0	0	0	2	3	0	22	23	4	0	2	3	1	**60**
	(0.00)	(0.00)	(0.00)	(0.00)	(0.00)	(0.00)	(0.60)	(1.07)	(0.00)	(6.02)	(5.37)	(0.99)	(0.00)	(0.51)	(0.67)	(0.43)	

## Data Availability

A.27 consensus genome sequences used in this study and obtained via DESH were uploaded to GISAID and the EMBL-EBI COVID-19 Data Portal (PRJEB44987). Note, that due to sequencing or reconstruction errors (e.g., causing frameshifts) not all A.27 genome sequences obtained via DESH could be uploaded to GISAID immediately. For example, EPI_ISL_1353586 could not be uploaded to GISAID due to an incorrectly masked deletion at position 28,252 causing a frameshift. Once potential errors in the sequences are fixed, which currently prevent uploading, the remaining sequences will also be made publicly available on GISAID. Meanwhile, all sequences and metadata obtained via DESH are uploaded daily to doi.org/10.5281/zenodo.5139363 (accessed on 27 July 2021) including also all A.27 sequences used in this study.

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
