# Peer review of "Rise and Fall of SARS-CoV-2 Lineage A.27 in Germany"

_viruses, 2021, doi:10.3390/v13081491_

Round 1
Reviewer 1 Report
Calvignac-Spencer et al. describes the emergence and decline of a unique SARS-CoV-2 mutant strain A.27 in the German state Baden-Wuerttemberg. This paper presents interesting work, and main conclusions are well supported by the data.
I have a few comments to make:
- Could authors shed light on the origin of A.27 lineage in discussion? These are locally derived mutants or imported from other countries (e.g. France). Interesting A.27 differs from other VOCs with D614G background, this suggests A.27 may evolve from 614D form (minority strains in Europe).
- Line 50, Denmark in mid-December, which year, 2020?
- Figure 1A, why Berlin and Bavaria curves have exact same color, should be different.
- Figure 2, can authors label MRCA date on the tree, or include in figure legend; also include MRCA standard deviation (date range) if available. In addition, it is hard to spot A.27 sequences from the tree, could authors make it more visible?
- Describe more on sequencing method, Artic amplicon sequencing protocol or others, how about sequencing depth or coverage cut-off for assembly
- Line 141, I agree with authors those three genomes from GISIAD need further confirmation of SNPs, due to presence of possible sequencing errors and coverage variations (may lead to loss of characteristic SNPs). D614G could also come from sequencing contamination, where was this genome sequenced (in-house or by other group)?
Author Response
> I have a few comments to make:
__RESPONSE:__ Thanks for your comments and suggestions!
> Could authors shed light on the origin of A.27 lineage in discussion? These are locally derived mutants or imported from other countries (e.g. France). Interesting A.27 differs from other VOCs with D614G background, this suggests A.27 may evolve from 614D form (minority strains in Europe).
__RESPONSE:__ This is indeed an interesting question we also already thought about and to our knowledge the origin of A.27 is still not fully solved and requires further investigation. We expanded the Conclusion accordingly and also add here some further thoughts.
We already mentioned two other studies [1,2] in the introduction that investigated characteristics of the A.27 lineage and also speculated a bit about its origin. In the first study from Colson et al., the authors describe eight sequences from Mayotte, a French overseas department in the Comoros archipelago, as well as others originated from France (n = 22), Denmark (n = 2), the Netherlands (n = 2), Belgium (n = 2), England (n = 1), Turkey (n = 4) and Nigeria (n = 1). The second study from Mallm et al. focuses more on sequences detected from the Rhine-Neckar district in Germany during January-March 2021. Here, the authors also show that A.27 is phylogenetically separated from other N501Y variants (B.1.1.7, P.1, B.1.351).
On top of these two studies, Anoh et al. [3] reported about genomic surveillance in Côte d'Ivoire (western sub-Saharan Africa) between May 2020 and March 2021. Here, the authors also found that ~20-30% of the cases in early 2021 were caused by A.27 (https://www.medrxiv.org/content/10.1101/2021.05.06.21256282v1). Their results further suggest that A.27 might have spread much further across the African continent. In addition, A.27 also reached pretty high levels in Burkina Faso (another place where the RKI supports genomic surveillance) and Togo (supported by a French network). However, nothing is published so far but sequences are available on GISAID. Thus, we downloaded sequences from GISAID (2021-07-20) and screened for
* sampling date starting from 1st December 2020
* and submissions from Africa
The latest GISAID-submitted sequences from Burkina Faso or Togo were from 2021-02-15 and 2021-03-20, respectively. We investigated, how many of these sequences can be assigned to A.27. For Burkina Faso, we found 170 sequences from which 8 % could be assigned to A.27. For Togo (n=96) 21 % could be assigned to A.27. All these A.27 sequences appear between the end of December 2020 and mid of February in Burkina Faso and Togo, which fits with our observations of A.27 occurrence in Germany if we assume an earlier origin in West Africa. We also found a higher percentage of A.27 sequences in Tunisia between February and March 2021 (12.96%, n=7 A.27 sequences). Thus, we speculate that West Africa may be the origin of the A.27 lineage, which may also explain why early and higher numbers of this specific lineage were discovered in France and Mayotte first. However, we must consider sparse sampling strategies and other caveats (random vs. suspect sampling, travel history, sequencing capacity, overall reporting, and data availability). Therefore, further analyses are needed to fully elucidate the origin of the SARS-CoV-2 A.27 lineage.
We added this investigation to the Conclusion and to path the way for further in-depth investigations of the origin of A.27.
We added a paragraph to the manuscript.
[1] Colson et al. "Spreading of a new SARS-CoV-2 N501Y spike variant in a new lineage". Clinical Microbiology and Infection 2021.
[2] Mallm et al. "Local emergence and decline of a SARS-CoV-2 variant with mutations L452R and N501Y in the spike protein". medRxiv 2021.
[3] Anoh, Etilé Augustin, et al. "SARS-CoV-2 variants of concern, variants of interest and lineage A. 27 are on the rise in Côte d'Ivoire." medRxiv (2021).
> Line 50, Denmark in mid-December, which year, 2020?
__RESPONSE:__ 2020. We added that, thanks for the comment.
> Figure 1A, why Berlin and Bavaria curves have exact same color, should be different.
__RESPONSE:__ Thanks, we corrected the figure accordingly.
> Figure 2, can authors label MRCA date on the tree, or include in figure legend; also include MRCA standard deviation (date range) if available. In addition, it is hard to spot A.27 sequences from the tree, could authors make it more visible?
__RESPONSE:__ Thanks for the suggestions, we improved the figure accordingly and also increased the size by moving the legend. Hopefully, the differences between the circles (A.27) and triangles (other A lineages) are now better visible. We also added the MRCA date to the caption and additionally marked it in the figure.
> Describe more on sequencing method, Artic amplicon sequencing protocol or others, how about sequencing depth or coverage cut-off for assembly
__RESPONSE:__ We added additional information on the sequencing metrics of the three in-house sequenced A.27 samples to the Methods.
> Line 141, I agree with authors those three genomes from GISIAD need further confirmation of SNPs, due to presence of possible sequencing errors and coverage variations (may lead to loss of characteristic SNPs). D614G could also come from sequencing contamination, where was this genome sequenced (in-house or by other group)?
__RESPONSE:__ All three genomes were sequenced by other groups thus we, unfortunately, don't have raw sequencing data available yet. You are right, D614G could also come from contamination/co-infection. So far, we, as the national public health institute, only get already reconstructed genome sequences which makes it, unfortunately, difficult/impossible to track such sequences down in more detail. Once we also get raw read data (people are working on a system so that labs can also submit raw reads) we also want to investigate such "strange" sequences in more detail and to confirm observed mutations and rule out sequencing/reconstruction errors.
Reviewer 2 Report
This original article should be completed in conclusion using more phylogenetic information.
Author Response
> This original article should be completed in conclusion using more phylogenetic information.
__RESPONSE:__ Thanks for this suggestion. We added a more detailed discussion/conclusion about the potential origin of the A.27 lineage, also including phylogenetic assessment from other A.27 studies and the most recent GISAID data from Africa. In addition, we are working on a follow-up study of this interesting lineage that will also include a more up-to-date data set and phylogenetic comparison. Together with other groups from France we also want to look into more detailed phylogenetic analyses (BEAST, ...) to elucidate the origin of A.27 in the future.